# Image Segmentation Approaches for Weld Pool Monitoring during Robotic Arc Welding

**Zhenzhou Wang \*, Cunshan Zhang, Zhen Pan, Zihao Wang, Lina Liu, Xiaomei Qi, Shuai Mao and Jinfeng Pan**

College of Electrical and Electronic Engineering, Shandong University of Technology, China;
zcs@sdut.edu.cn (C.Z.); pz15615732790@163.com (Z.P.); 13110473098@stumail.sdut.edu.cn (Z.W.);
linaliu-126@163.com (L.L.); qixiaomei@sdut.edu.cn (X.Q.); maoshuai04965@sdut.edu.cn (S.M.);
pjfbysj@163.com (J.P.)
**\*** Correspondence: wangzz@sdut.edu.cn

**Abstract:** There is a strong correlation between the geometry of the weld pool surface and the degree of penetration in arc welding. To measure the geometry of the weld pool surface robustly, many structured light laser line based monitoring systems have been proposed in recent years. The geometry of the specular weld pool could be computed from the reflected laser lines based on different principles. The prerequisite of accurate computation of the weld pool surface is to segment the reflected laser lines robustly and efficiently. To find the most effective segmentation solutions for the images captured with different welding parameters, different image processing algorithms are combined to form eight approaches and these approaches are compared both qualitatively and quantitatively in this paper. In particular, the gradient detection filter, the difference method and the GLCM (grey level co-occurrence matrix) are used to remove the uneven background. The spline fitting enhancement method is used to remove the fuzziness. The slope difference distribution-based threshold selection method is used to segment the laser lines from the background. Both qualitative and quantitative experiments are conducted to evaluate the accuracy and the efficiency of the proposed approaches extensively.

**Keywords:** Image processing; segmentation; spline; grey level co-occurrence matrix; gradient detection; threshold selection

## 1. Introduction

Arc welding is a widely used process for joining various metals. The electric current is transferred from the electrode to the work piece through the arc plasma, which has been modeled to control the weld quality in the past research [1,2]. However, the direct factor that affects the weld quality is the geometry of the weld pool instead of arc plasma because the skilled welders achieve good weld quality mainly based on the visual information of the weld pool. During arc welding, the incomplete weld pool penetration will reduce the effective working cross-sectional area of the weld bead and subsequently reduce the weld joint strength. It also causes stress concentrations in some cases, e.g., the fillet and T-joints. On the contrary, excessive weld pool penetration might cause melt-through. The skilled welder needs to adjust the position and travelling speed of the weld torch based on the information of the observed weld pool surface to achieve complete penetration. The shortage of skilled welders and a need for welds of a consistently high quality fuels an increasing demand for automated arc welding systems. It is believed that machine vision techniques will lead the development of the next generation intelligent automated arc welding systems.

In recent years, great efforts have been put to develop the automated and high-precision welding equipment to achieve high quality welded joints consistently. The important parts of this kind of automated welding equipment include the seam tracking system [3–6], the weld penetration monitoring system [7–9] and the control system. Desirably, the seam tracking system or the monitoring system and control system are in a closed loop. Thus, the tracking result or the monitoring result could serve as feedback for the control system to ensure high quality of welding. In the past studies, it has been shown that weld pool surface depression has a major effect on weld penetration [10–14]. The geometry of the weld pool surface will affect the convection in the pool. The primary welding parameter, the welding current density is also affected by the geometry of the weld pool greatly. In return, the plasma and the welding current affect the geometry of the weld pool. Both the arc plasma and the molten weld pool are affected by the current density distribution. Therefore, it is fundamental for the automated welding systems to take quantitative measurements of the weld pool surface.

In the past decades, a lot of research has been conducted to measure the shape of the arc welding weld pool by structured light methods [9,15–23]. In Reference [15], the authors pioneered to measure the deformation of the weld pool by inventing a novel sensing system. Their sensing system projects a short duration pulsed laser light through a frosted glass with a grid onto the specular weld pool. The reflected laser stripes are imaged in a CCD camera and the geometry of the reflected stripes contain the weld pool surface information. This method might be the earliest method that made good use of the reflective property of the weld pool surface and achieved state of the art accuracy at that time. In Reference [16], the authors improved the structured light method for gas tungsten arc weld (GTAW) pool shape measurement by reflecting the structured light onto an imaging plane instead of onto the image plane of the camera directly. The imaging plane is placed at a properly selected distance. Thus, the imaged laser patterns are clear enough while the effect of the arc plasma has attenuated significantly, because the propagation of the laser light is much longer than that of the arc plasma. Ever since then, this weld pool sensing technique has become the mainstream of weld pool imaging technology [17–23]. In References [17,18], the authors tried to increase the accuracy of measuring the 3D shape of the GTAW weld pool sensed by the same imaging system as [16]. In Reference [19], the authors came up with an approach for segmentation of the reflected laser lines for pulsed gas metal arc weld (GMAW-P). In References [9,20], the reflected laser lines from the GMAW-P weld pool were segmented manually to measure the weld pool oscillation frequency. In Reference [21], two cameras were used to measure the shape of the weld pool for GMAW-P and an unsupervised approach was proposed to segment and cluster the reflected laser lines. In Reference [22], three cameras are used to measure the specular shape from the projected laser rays and an unsupervised approach was proposed to reconstruct the GTAW weld pool shape with closed form solutions. It achieved on line robust measurement of GTAW weld pool shape with three calibrated cameras.

There is one major difference for the used laser pattern among these structured light methods [9,15–23]. The laser dot pattern is used for the measurement of the GTAW weld pool surface while the laser line pattern is used for the measurement of the GMAW weld pool surface. Compared to the GTAW weld pool surface, the GMAW weld pool surface is much more dynamic and fluctuating, because GMAW process transfers additional metallic and liquid droplets into the weld pool, which increases the fluctuation and dynamics of the pool surface greatly. The position and geometry of the local specular surface changes rapidly and greatly, which causes the reflected rays to change their trajectories rapidly and greatly. If the laser dot pattern is used, the reflected laser dots might interlace irregularly, which makes it impossible for the unsupervised clustering method to identify these dots robustly. Therefore, the laser line patterns are used in measuring the weld pool surface of the GMAW process [19–21]. Robust segmentation and clustering of the reflected laser lines become the most important and challenging part in the whole monitoring system, because of the uncertainty of the weld pool geometry. The quality of the captured image is significantly affected by the welding parameters. Due to the lack of generality and robustness, the proposed image processing methods might work for the images captured in some specifically designed welding experiments,

while might not work for the images captured in other experiments. For instance, the reflected laser lines were filtered by the top-hat transform and then segmented by thresholding in Reference [19]. It yields acceptable segmentation results in many cases. However, it fails completely when it is used to segment the images captured in Reference [21] with different welding parameters. The reflected laser lines were filtered by the fast Fourier transform (FFT) and then segmented by a manually specified threshold in Reference [20]. It also fails completely in segmenting the images shown in Reference [21]. To segment the reflected laser lines more robustly, a difference method and an effective threshold selection method are proposed in Reference [21]. The segmented laser lines were then clustered based on their slopes. One big problem that could be seen from the experimental figures in References [19–21] is that quite a few of the laser line parts are missing, because of the limitations of their proposed image processing methods. One direct consequence of missing a significant part of the reflected laser line is the inaccurate characterization of the weld pool shape, since the length of the reflected laser is proportional to the size of the weld pool. Another drawback is that the segmented small part of the laser line might be deleted as noise blobs or some large noise blobs might be recognized as part of the reflected laser line during the unsupervised clustering. As a result, the laser line might be clustered incorrectly.

In Reference [23], a new approach was proposed, and it achieved significantly better segmentation accuracy compared to the past research [19–21]. It comprises several novel image processing algorithms: A difference operation, a two-dimensional spline fitting enhancement operation, a gradient feature detection filter and the slope difference distribution-based threshold selection. The major goal of [23] is to cluster and characterize the laser lines under extremely harsh welding conditions. As a result, it omits the image processing methods for the images captured under less harsh welding conditions with mild welding parameters. In addition, quantitative results and comparisons to validate the effectiveness of the proposed segmentation approach were also not given. Due to the page limit, the reasons why the segmentation approach should contain these image processing algorithms were not explained adequately. One goal of this paper is to complement the research work conducted in Reference [23] and conduct a more thorough experiment to find the most effective solution both qualitatively and quantitatively. Another goal is to come up with more efficient segmentation approaches for images with relatively high quality that are captured under mild welding parameters.

In the past research, the visual inspection of the weld pool surface was mainly used to understand the complex arc welding processes. The obtained data were used to validate and improve the accuracy of the numerical models and to gain insight into the complex arc welding processes. Few are used for in-process welding parameter adjustment and on-line feedback control, due to the lack of a robust and efficient approach that is capable of extracting meaningful feedback information on-line from most of the captured images. The proposed approaches in this paper are promising to accomplish this challenging task in the future.

This paper is organized as follows. Section 2 describes the established monitoring system. In Section 3, state of the art methods for laser line segmentation are evaluated and compared. In Section 4, the combination approach proposed previously is explained theoretically. In Section 5, we propose different segmentation approaches by combining different image processing algorithms. In Section 6, the experimental results and discussions are given. Section 7 concludes the paper.

## 2. The Structured Light Monitoring System

Figure 1 shows the configuration of the popular weld pool monitoring system that has been adopted in References [9,15–23]. The major parts of the system include two point grey cameras, $C1$ and $C2$, and one Lasiris SNF with the wavelength at 635 nm. A linear glass polarizing filter with the wavelength from 400 nm to 700 nm is mounted on camera $C2$ to remove the strong arc light. The structured light laser line pattern is projected by the Lasiris SNF laser generator onto the weld pool surface and reflected onto the diffusive imaging plane $P1$. The calibrated camera $C2$ views the reflected laser lines from the back side of $P1$, which is made up of a piece of glass and a piece

of high-quality paper. To facilitate the computation, $P1$, the $YZ$ plane and the image plane of the calibrated camera $C2$ are set to be parallel to each other. During calibration, the laser lines are projected by the Lasiris SNF laser onto a horizontal diffusive plane and camera $C1$ is used to calculate the length of the straight laser lines in the world coordinate system. Then the horizontal diffusive plane is replaced by a horizontal mirror plane, which reflects the laser lines onto the vertical diffusive plane $P1$. The calibrated camera $C2$ is used to calculate the length of the imaged straight laser lines in the world coordinate system. Both camera $C1$ and camera $C2$ are not attached to the welding platform to avoid the vibration generated during the welding process.

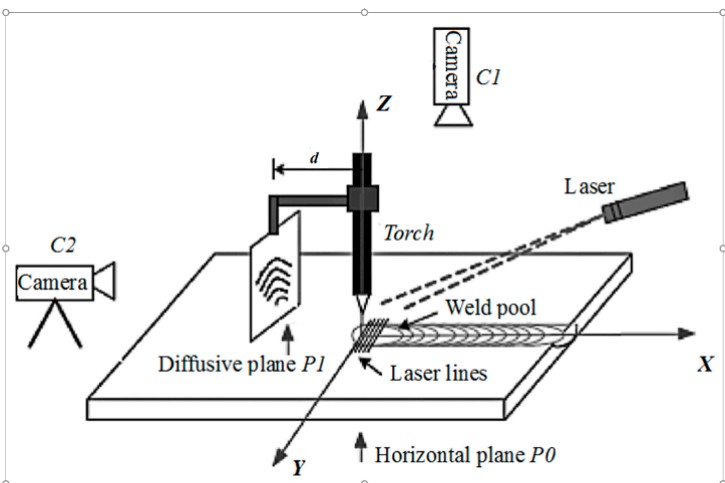

**Figure 1.** The configuration of the weld pool monitoring system.

## 3. State of the Art Methods for Laser Line Segmentation

### 3.1. Top-Hat Transform Based Method

In Reference [19], the uneven background caused by the arc light was removed by the top-hat transform, which is formulated as [24]:

$$T(f) = f - f \cdot e, \tag{1}$$

where $\cdot$ denotes the opening operation, $f$ denotes the image and $e$ denotes the structuring element. $e$ is chosen as a disk with radius 15 in this research work. After the image was enhanced by the top-hat transform, the image was segmented by the threshold selection method. The selected welding parameters are as follows. The wire feed speed is 55 mm/s, the welding speed is 5 mm/s, the peak current is 220 A and the base current is 50 A. As a result, the captured image is very clear for segmentation. However, their segmentation result still misses significant parts for the laser lines.

### 3.2. FFT Filtering Based Method

In Reference [20], fast Fourier transform [25] was used to remove the uneven arc light caused background and then the image was segmented by threshold selection. The selected welding parameters are as follows. The welding speed is 0 mm/s, the peak current is 160 A and the base current is 80 A. With these welding parameters, the change of the weld pool's geometry is relatively slow compared to that in Reference [19]. As a result, the captured images are relatively easier for automatic image processing. Hence, the authors could segment the captured images by specifying the threshold manually after FFT filtering.

### 3.3. Difference Operation Based Method

In Reference [21], the difference method was proposed to reduce the unevenly distributed background. The differenced image is obtained by the following operation.

$$f_d(x,y) = f(x + \Delta d, y) - f(x,y), \tag{2}$$

where $\Delta d$ denotes the step size of the operation and it is determined by off line analysis of the average width of the laser lines. The step size should be greater than or equal to the width of the laser line in the captured images and it is selected as 10 in this research work. The laser lines are then segmented from the differenced image by the threshold selection method.

### 3.4. Grey Level Co-Occurrence Matrix Based Method

Grey level co-occurrence matrix (GLCM) [26] computes the frequencies of different combinations of pixel values or grey levels occurring in an image and then forms a matrix to represent these frequencies. Therefore, it is usually used to segment textured images or objects with an unevenly distributed background. The second order GLCM is usually used for segmentation and it is formulated as:

$$P(i,j|d,\theta) = \frac{\#\{k,l \in D | f(k) = i, f(l) = j, \| k - l \| = d, \angle(k - l) = \theta\}}{\#\{m,n \in D | \| m - n \| = d, \angle(m - n) = \theta\}}, \tag{3}$$

where $d$ is the distance of pixel $k$ and pixel $l$. $\theta$ is the angle of vector $(k - l)$ with the horizontal line or vertical line. The combination of $d$ and $\theta$ represents the relative position of pixel $k$ and pixel $l$, which have gray-scale value $i$ and $j$ respectively.

During our implementation, we update all the intensity values in the moving window by subtracting its minimal intensity value and then adding one. The quantization level of the GLCM matrix becomes one and the maximal intensity value of the updated moving window. Following the computation of the GLCM, the contrast measure is used to form the GLCM image. Then, the GLCM image is segmented by a global threshold.

### 3.5. Combination Method

In Reference [23], a combination approach was used to segment the laser lines. The combined image processing algorithms include a difference operation, a two-dimensional spline fitting enhancement operation, a gradient feature detection filter and the slope difference distribution-based threshold selection. In both [21,23], the laser lines are segmented by the slope difference distribution-based threshold selection method that could be summarized as follows. Firstly, the gray-scales of the original image is rearranged in the interval from 1 to 255 and its normalized histogram distribution $P(x)$ is computed. Secondly, the normalized histogram $P(x)$ is smoothed by the fast Fourier transform (FFT) [25] based low pass filter with the bandwidth $W = 10$. Thirdly, two slopes, the right slope and the left slope, are computed for each point on the smoothed histogram distribution. The slope difference distribution is computed as the differences between the right slopes and their corresponding left slopes. In the slope difference distribution, the position where the valley with the maximum absolute value occurs is selected as the threshold. The slope difference distribution-based threshold selection method is critical in this application. For the comparison with state of the art image segmentation methods, please refer to the related research work [21], where this threshold selection method is compared with the state of the art method using the same type of images. The comparisons with state of the art image segmentation methods showed that the slope difference distribution based threshold selection method is significantly more accurate in segmenting some specific types of images, including the laser line images.

*3.6. Performance Evaluation*

Figure 2a shows a typical image captured by the structured light monitoring system with the following welding parameters. The wire feed speed is 84.67 mm/s, the welding speed is 3.33 mm/s, the peak current is 230 A and the base current is 70 A. As can be seen, these parameters are significantly higher than those used in References [19,20]. With these parameters, the resultant weld pool surface changes more rapidly and irregularly. Hence, the quality of the captured image in this study is much reduced. The results by top hat method, FFT method, difference method, GLCM method and combination method are shown in Figure 2b–f respectively. Compared to the segmentation result by combination method, the segmentation results by other state of the art methods appear to be very inaccurate. Although the difference method achieves the second best result, there is one big segmented line on the top caused by the edge of the imaging plane. It costs additional effort for the subsequent unsupervised clustering. In addition, there are noise blobs that are hard to be distinguished from the laser line parts.

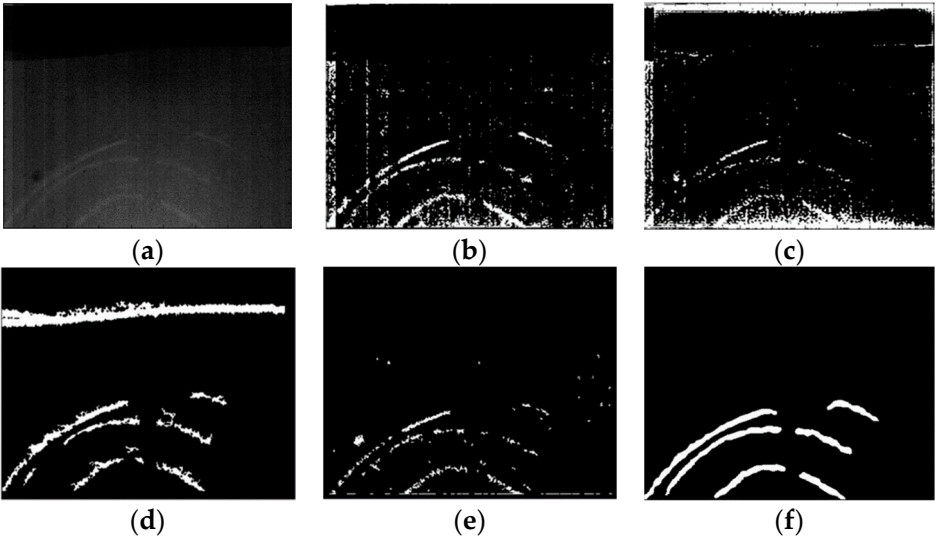

**Figure 2.** Evaluation of state of the art methods (**a**) one typical captured image; (**b**) segmentation result by the top hat method; (**c**) segmentation result by the FFT method; (**d**) segmentation result by the difference method; (**e**) segmentation result by the GLCM method; (**f**) segmentation result by combination method.

## 4. Analysis of the Combination Approach

In Reference [23], a combination approach is proposed to segment the reflected laser lines as accurate as possible. The flowchart of this segmentation approach comprises a difference operation, a two-dimensional spline fitting enhancement operation, a gradient feature detection filter and the slope difference distribution-based threshold selection. However, the reasons why the segmentation approach should contain these image processing algorithms were not explained adequately in Reference [23]. Quantitative results to validate the effectiveness of this segmentation approach were also not given. Here, we will theoretically explain why the proposed two-dimensional spline fitting enhancement method and the gradient feature detection filter work well in segmenting the laser lines.

The intensity distribution, $I_a$ of the captured image caused by the arc light could be modeled mathematically by the following equation [21]:

$$I_a(T) = \tau \left| \frac{2hv^3}{c^2} \times \frac{\vec{N} \times cos\beta \times O}{e^{\frac{hv}{\gamma T}} - 1} \right| \times \frac{d}{r^3} = \frac{C_a(T)}{r^3}, \tag{4}$$

where $\tau$ is the intensity mapping function of the CCD camera [27]. $\vec{N}$ is the surface normal at position $p$. $\beta$ is the angle between the surface normal and the incident light. $O$ denotes the color value. $r$ is the distance between the arc light center and position $p$. $d$ is the distance from the arc light center to the diffusive imaging plane $P1$. $v$ is the spectral frequency of arc light and $h$ is the planck's constant. $c$ is the speed of the light and $\gamma$ is the boltzmann's constant. $T$ is the temperature of arc light source, which is determined by the welding current that alternates with the frequency of 10 Hz. The frame rate of the camera $C2$ is set to 300 frames per second during the experiment. Hence, 30 images are captured with different sampled currents at one period of the current wave. The temperature $T$ is determined by the value of the current and thus it changes with the current from frame to frame in the time domain. In the same frame, the temperature $T$ is a constant. At each image point, both the intensity mapping function $\tau$ and the angle $\alpha$ remain the same in all the captured image sequences. Thus, $C^a$ is a constant at each specific image point in one frame and it varies from frame to frame with the value of $T$. From Equation (4), it becomes much obvious that the intensity distribution produced by the arc light effect on the captured image is inversely proportional to the $r^3$. Thus, the intensity distribution caused by the arc light can be modeled as:

$$a(x,y) = \frac{C_a(T)}{\left[(x-x_0)^2 + (y-y_0)^2 + d^2\right]^{3/2}},$$

(5)

where $(x_0, y_0)$ denotes the center of arc light distribution and it may lie outside of the image. The arc light is affected by the additional laser line and can be modeled as:

$$f(x,y) = l(x,y) + a(x,y),$$

(6)

where $l(x,y)$ denotes the intensity distribution of the laser line and it is formulated as:

$$l(x,y) = \begin{cases} w\mu(x,y); & (x,y) \in A \\ 0; & (x,y) \notin A \end{cases},$$

(7)

where A denotes the laser line area, $w$ is a constant whose value is higher than the average value of the arc light distribution, $I_a$. $\mu(x,y)$ is the membership function that represents the fuzziness of the laser line and is formulated as:

$$\mu(x,y) = Exp\left(-\frac{(L(x,y) - \mu_L)^2}{2\sigma_L{}^2}\right),$$

(8)

where $L(x,y)$ denotes the ideal intensity distribution of the laser line. $\mu_L$ is its mean and $\sigma_L$ is its variance respectively.

Combining Equations (5)–(8), we get the model of the intensity distribution for the image with reflected laser lines.

$$f(x,y) = \begin{cases} w\mu(x,y) + \frac{C_a(T)}{\left[(x-x_0)^2 + (y-y_0)^2 + d^2\right]^{3/2}}; & (x,y) \in A \\ \frac{C_a(T)}{\left[(x-x_0)^2 + (y-y_0)^2 + d^2\right]^{3/2}}; & (x,y) \notin A \end{cases}.$$

(9)

From the above model, we see that the fuzziness caused by $\mu(x,y)$ should be reduced effectively to obtain high segmentation accuracy. However, the moving average filter can only reduce the Gaussian noise instead of removing the fuzziness. Thus, a new enhancement method is required.

After the image fuzziness has been added in the derived image model (Equation (9)), we need to theoretically find the reasons that why the fuzziness causes parts of the segmented laser lines missing. To this end, the mathematical explanation of how the gradient detection filter works for segmenting

the objects from the uneven background is described at first. According to Equation (9), the gradient value at the position $(x, y)$ that is caused by the arc light background can be formulated as:

$$g^a = \frac{C_a(T)}{\left[\left(\frac{N-1}{2}+x-x_0\right)^2+\left(\frac{N-1}{2}+y-y_0\right)^2+d^2\right]^{3/2}} - \frac{C_a(T)}{\left[\left(x-x_0-\frac{N-1}{2}\right)^2+\left(y-y_0-\frac{N-1}{2}\right)^2+d^2\right]^{\frac{3}{2}}}. \tag{10}$$

The gradient values caused by the laser line can be formulated as:

$$g^l = \begin{cases} w\mu(x,y)+g^a; & if\ top\ part\ of\ K_g \notin A \\ -w\mu(x,y)+g^a & if\ bottom\ part\ of\ K_g \notin A \\ g^a; & if\ whole\ part\ of\ K_g \in A \end{cases}. \tag{11}$$

For the designed filter $K_g(N, \theta)$ in this research work, the following conditions need to be met.

$$w\mu(x,y) \gg g^a. \tag{12}$$

Since $g^a$ is the gradient of a small part of the background with a range of $N$, its value is much reduced compared to the variation of the total background. Thus, the condition of Equation (12) can be easily satisfied if $\mu$ is a constant to make the laser line gradient distinguished from the background variation. However, $\mu$ is the membership function and its value is random between 0 and 1. It is impossible to make Equation (12) true for every point. Hence, if the local fuzzy membership $\mu(x, y)$ could be fitted as a global function for the whole image to remove the randomness, Equation (12) could be easily satisfied. A two-dimensional spline function is an ideal global function to the image and thus it is used. The fitting process is implemented by minimizing the following energy function to get an enhanced image, $s(x, y)$ from the original image $f(x, y)$.

$$E = \frac{1}{2} \iint (s(x,y)-f(x,y))^2 dxdy + \frac{1}{2}\iint \left|\frac{d^2 s(x,y)}{dxdy}\right|^2 dxdy. \tag{13}$$

## 5. The Proposed Approaches

Although the same weld pool monitoring system has been adopted in References [9,15–23], the monitoring algorithms have been proposed differently and divergently. In addition, most proposed monitoring approaches only work for the specifically designed welding scenario. As described in Section 3, either the proposed monitoring approach in Reference [19] or the proposed monitoring approach in Reference [20] could not work for the weld pool scenario in Reference [23]. On the contrary, the proposed monitoring approach in Reference [23] works better for the weld pool scenario in both [19,20] than their proposed monitoring approaches. The reason lies in that the quality of the captured laser line images in References [19,20] is much higher than that of the captured images in Reference [23]. When the quality of the captured image is reduced greatly, the requirement for the monitoring algorithms increases significantly. The proposed monitoring approach is the most robust one for the structured laser line based weld pool monitoring systems shown in Figure 1 up to date. However, it might be redundant for monitoring of simple weld pool scenarios as described in References [19,20]. In Reference [19], GTAW process is used. The wire feed speed is 55 mm/s, the welding speed is 5 mm/s, the peak current is 220A and the base current is 50 A. In Reference [20], GMAW-P process is used. The welding speed is 0 mm/s, the peak current is 160 A and the base current is 80 A. In Reference [23], GMAW-P process is used. The wire feed speed is 84.67 mm/s, the welding speed is 3.33 mm/s, the peak current is 230A and the base current is 70 A. As a result, the produced weld pools in References [19,20] are much more stable than that produced in Reference [23]. Thus, the quality of the captured images in References [19,20] is much better than that of the images captured in Reference [23]. On the other hand, the processing time is also important for on line monitoring. Therefore, the combination approach proposed in Reference [23] is not always optimum when both

accuracy and efficiency are considered. The monitoring approach should be designed as fast as possible after the accuracy has been met.

In this paper, we combine the previously proposed image processing algorithms in different groups and form different segmentation approaches. We then evaluate their segmentation accuracy and segmentation efficiency quantitatively. Besides the five image processing algorithms proposed recently, we also add the traditional GLCM as an additional component. We name the segmentation by combining the difference method and threshold selection as *approach 1*, the segmentation by combining GLCM and threshold selection as *approach 2*, the segmentation by combining GLCM, the difference method and threshold selection as *approach 3*, the segmentation by combining the gradient detection filter and threshold selection as *approach 4*, the segmentation by combining the gradient detection filter, the difference method and threshold selection as *approach 5*, the segmentation by combining the gradient detection filter, spline fitting and threshold selection as *approach 6*, the segmentation by combining GLCM, spline fitting, the difference method and threshold selection as *approach 7*, the segmentation by combining the gradient detection filter, spline fitting, the difference method and threshold selection as *approach 8*.

As can be seen, there are five basic image processing methods that consist of (1), the difference method; (2), the GLCM; (3) the spline fitting; (4) the gradient detection filter; and (5), the threshold selection method. The difference method, the GLCM and the spline fitting have been explained by Equations (2), (3) and (13) respectively.

The gradient feature detection filter is formulated as:

$$K_g = R(VH, \theta), \tag{14}$$

where

$$V = [-k; v_1; v_2; \ldots; v_{N-1}; k], \tag{15}$$

$$H = [h_0, h_1, \ldots, h_{N-1}, h_N], \tag{16}$$

$$h_i = w_h(i); i = 0, \ldots, N, \tag{17}$$

$$v_i = w_v(i); i = 1, \ldots, N-1, \tag{18}$$

where $N$ equals the width of the laser line and determines the size of the kernel. $k$ is a constant. $w_h$ and $w_v$ are two weighting functions. As can be seen, the product $VH$ is a $N$ by $N$ matrix. $R(VH, \theta)$ is to rotate the matrix $VH$ by $\theta$ degrees in the counterclockwise direction around its center point. $\theta$ is orthogonal to the line direction and is chosen as $90°$ in this research.

The threshold selection method is implemented as follows. The histogram distribution of the image is computed, normalized and filtered by a low pass Discrete Fourier Transform (DFT) filter with the bandwidth 10. A line model is fitted with $N$ adjacent points at each side of the sampled point. The slopes of the fitted lines at point $i$, $a_1(i)$ and $a_2(i)$, are then obtained. The slope difference, $s(i)$, at point $i$ is computed as:

$$s(i) = a_2(i) - a_1(i); i = 16, \ldots, 240, \tag{19}$$

The continuous function of the above discrete function, $s(i)$ is the slope difference distribution, $s(x)$. To find the candidate threshold points, the derivative of $s(x)$ is set to zero.

$$\frac{ds(x)}{dx} = 0, \tag{20}$$

Solving the above equation, the valleys $V_i$; $i = 1, \ldots, N_v$ of the slope difference distribution are obtained. The position where the valley $V_i$ yields the maximum absolute value is chosen as the optimum threshold in this specific application.

## 6. Results and Discussion

### 6.1. Experimental Results

To rank the accuracy and the efficiency the proposed approaches with different algorithm combinations, we show the qualitative results in Figures 3–12 for visual comparison. As can be seen, the performances of the proposed approaches vary depending on the quality of the captured images. For some images (e.g., Figures 11 and 12), most approaches work well. Considering the computation time, *approach 8* is not always the optimum solution in monitoring different welding processes with different welding parameters.

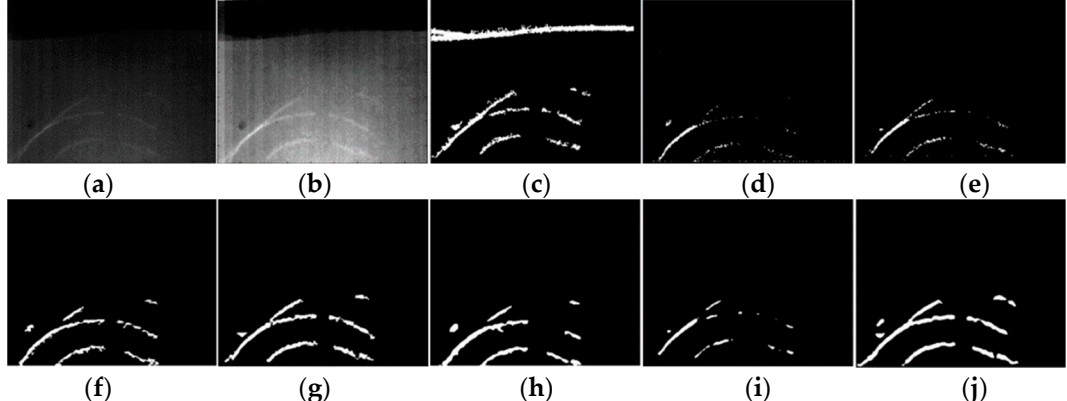

**Figure 3.** Performance comparison of the eight methods with image 1; (**a**) original image; (**b**) filtered image; (**c**) *approach 1*; (**d**) *approach 2*; (**e**) *approach 3*; (**f**) *approach 4*; (**g**) *approach 5*; (**h**) *approach 6*; (**i**) *approach 7*; (**j**) *approach 8*.

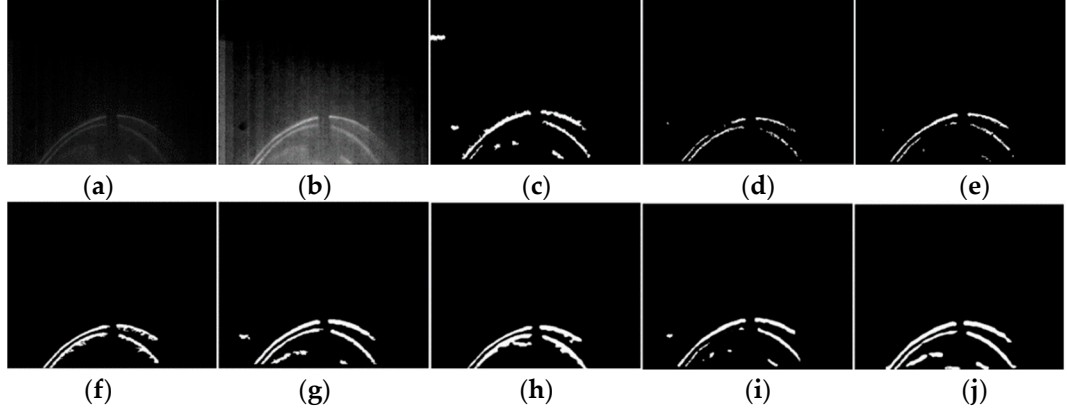

**Figure 4.** Performance comparison of the eight methods with image 2; (**a**) original image; (**b**) filtered image; (**c**) *approach 1*; (**d**) *approach 2*; (**e**) *approach 3*; (**f**) *approach 4*; (**g**) *approach 5*; (**h**) *approach 6*; (**i**) *approach 7*; (**j**) *approach 8*.

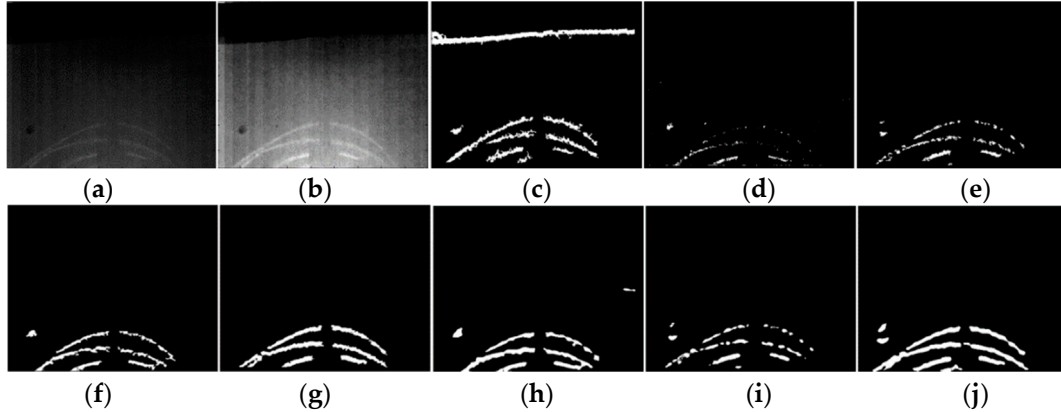

**Figure 5.** Performance comparison of the eight methods with image 3; (**a**) original image; (**b**) filtered image; (**c**) *approach 1*; (**d**) *approach 2*; (**e**) *approach 3*; (**f**) *approach 4*; (**g**) *approach 5*; (**h**) *approach 6*; (**i**) *approach 7*; (**j**) *approach 8*.

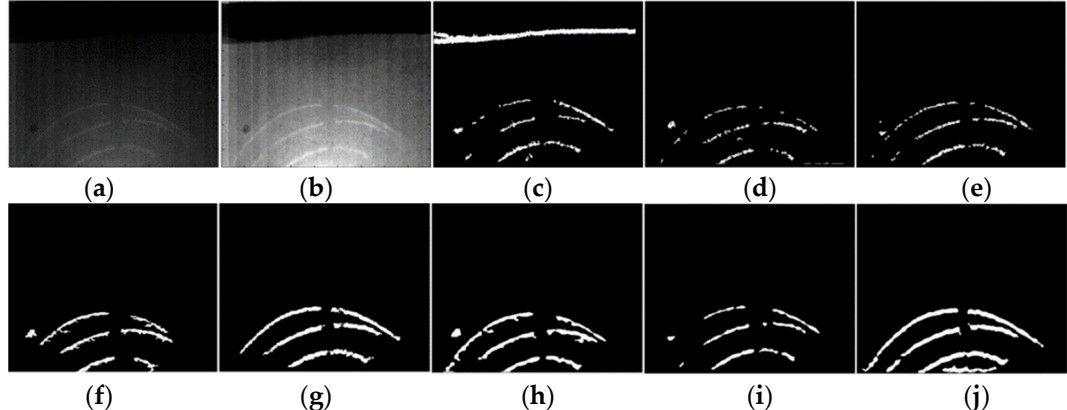

**Figure 6.** Performance comparison of the eight methods with image 4; (**a**) original image; (**b**) filtered image; (**c**) *approach 1*; (**d**) *approach 2*; (**e**) *approach 3*; (**f**) *approach 4*; (**g**) *approach 5*; (**h**) *approach 6*; (**i**) *approach 7*; (**j**) *approach 8*.

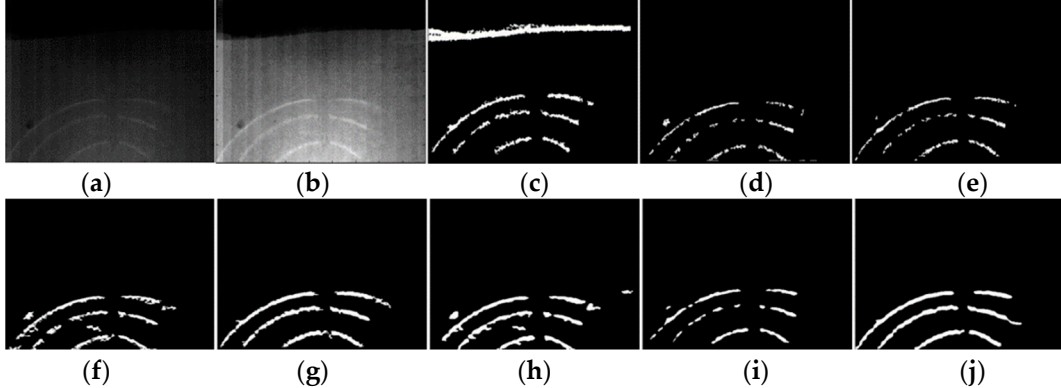

**Figure 7.** Performance comparison of the eight methods with image 5; (**a**) original image; (**b**) filtered image; (**c**) *approach 1*; (**d**) *approach 2*; (**e**) *approach 3*; (**f**) *approach 4*; (**g**) *approach 5*; (**h**) *approach 6*; (**i**) *approach 7*; (**j**) *approach 8*.

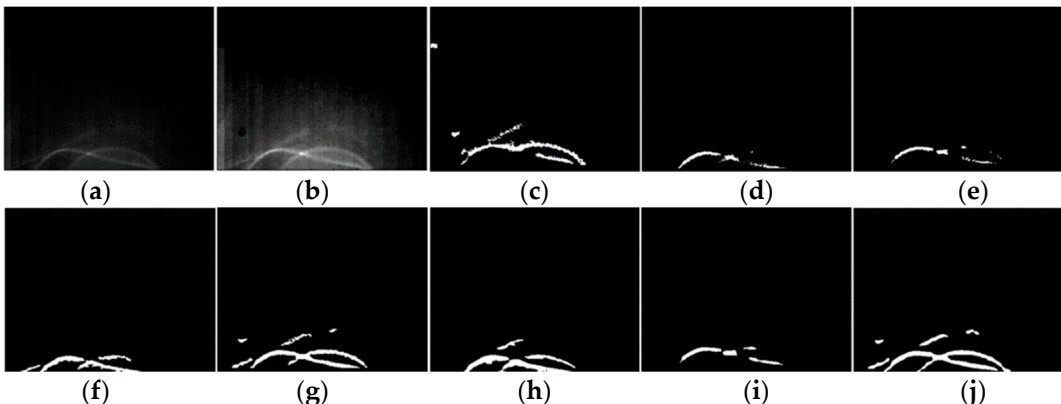

**Figure 8.** Performance comparison of the eight methods with image 6; (**a**) original image; (**b**) filtered image; (**c**) *approach 1*; (**d**) *approach 2*; (**e**) *approach 3*; (**f**) *approach 4*; (**g**) *approach 5*; (**h**) *approach 6*; (**i**) *approach 7*; (**j**) *approach 8*.

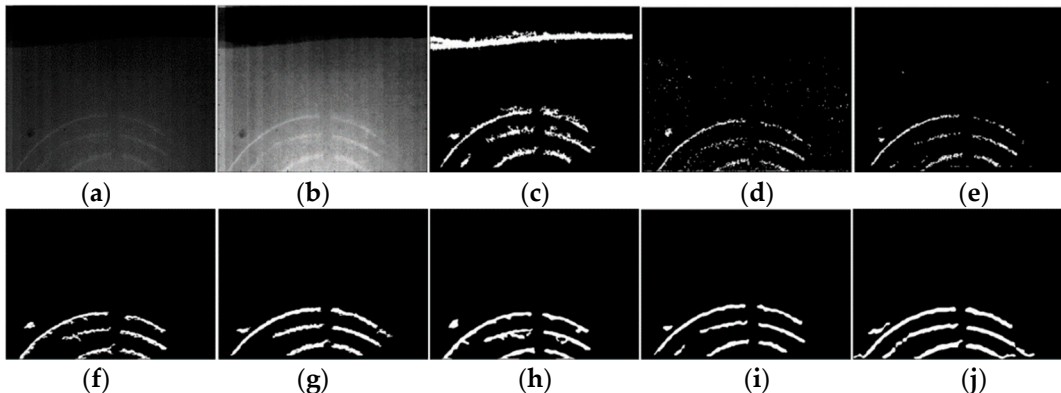

**Figure 9.** Performance comparison of the eight methods with image 7; (**a**) original image; (**b**) filtered image; (**c**) *approach 1*; (**d**) *approach 2*; (**e**) *approach 3*; (**f**) *approach 4*; (**g**) *approach 5*; (**h**) *approach 6*; (**i**) *approach 7*; (**j**) *approach 8*.

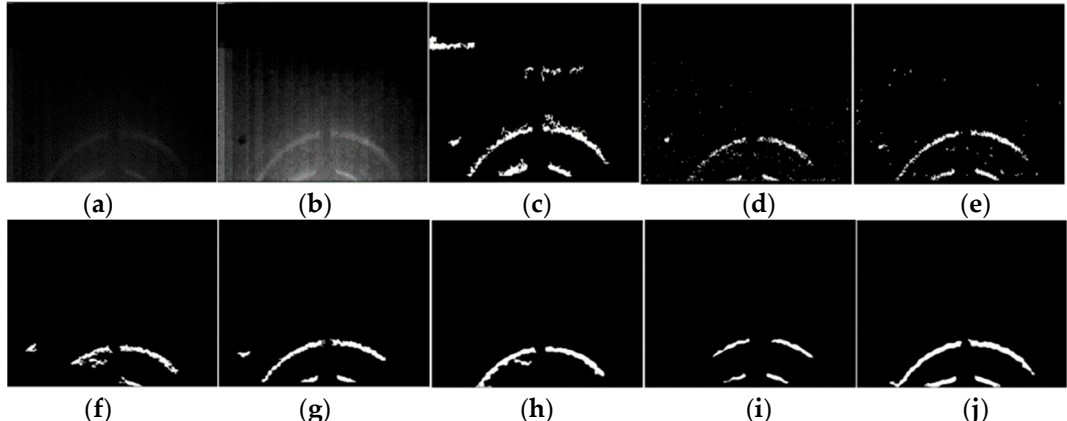

**Figure 10.** Performance comparison of the eight methods with image 8; (**a**) original image; (**b**) filtered image; (**c**) *approach 1*; (**d**) *approach 2*; (**e**) *approach 3*; (**f**) *approach 4*; (**g**) *approach 5*; (**h**) *approach 6*; (**i**) *approach 7*; (**j**) *approach 8*.

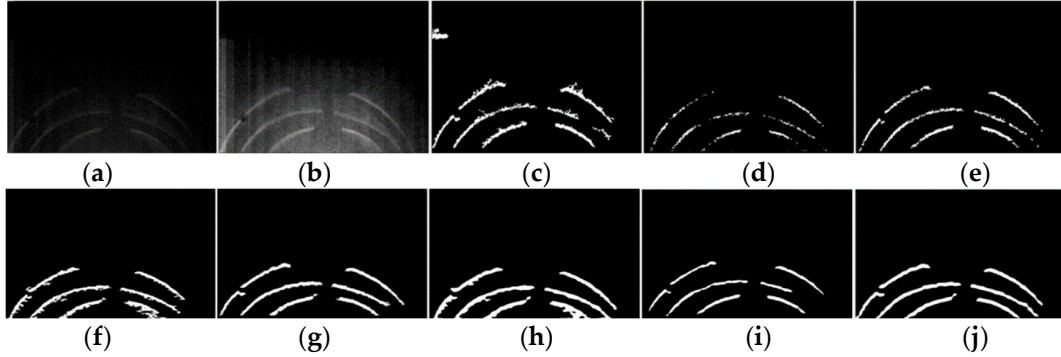

**Figure 11.** Performance comparison of the eight methods with image 9; (**a**) original image; (**b**) filtered image; (**c**) *approach 1*; (**d**) *approach 2*; (**e**) *approach 3*; (**f**) *approach 4*; (**g**) *approach 5*; (**h**) *approach 6*; (**i**) *approach 7*; (**j**) *approach 8*.

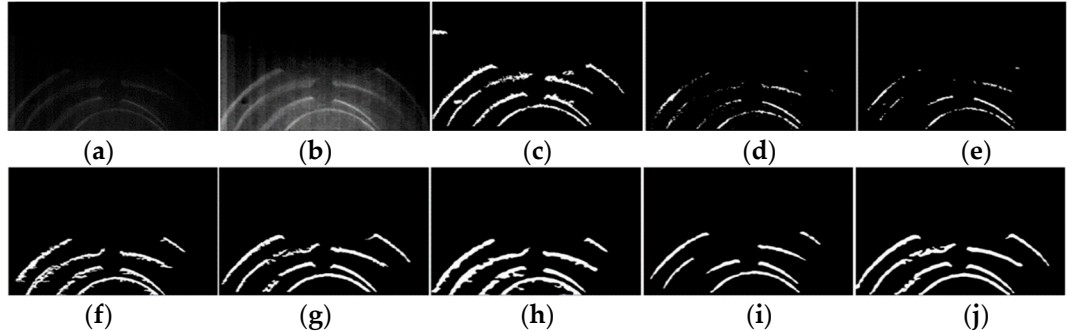

**Figure 12.** Performance comparison of the eight methods with image 10; (**a**) original image; (**b**) filtered image; (**c**) *approach 1*; (**d**) *approach 2*; (**e**) *approach 3*; (**f**) *approach 4*; (**g**) *approach 5*; (**h**) *approach 6*; (**i**) *approach 7*; (**j**) *approach 8*.

We use 30 images to compare the accuracy of these eight approaches quantitatively and the comparison is shown in Table 1. As can be seen, *approach 8* achieves the best segmentation accuracy. Since the computation time is also critical for on line monitoring, we compare the average computation times of processing the images by these eight approaches programmed with VC++ and Matrox image processing library on the computer with Intel i7-3770 3.4 GHz dualcore CPU. The comparison is shown in Table 2. As can be seen, *approach 1* using the difference operation proposed in Reference [21] is fastest while its segmentation accuracy is not acceptable for some images, e.g., Figure 8. The second fastest is *approach 4* using the gradient detection filter proposed in this paper and its segmentation accuracy is better than *approach 1*. The third fastest is *approach 5,* which combines the gradient dection filter and the difference operation and it achieves adequate segmentation accuracy for subsequent unsupervised processing. On the other hand, *approach 5* has achieved the second-best segmentation accuracy. Hence, *approach 5* is the optimum method for on line processing when the requirement for processing time is strict. In summary, *approach 8* is the best choice to segment the reflected laser lines with the highest accuracy for the developed GMAW weld pool monitoring system in Reference [23] while *approach 1* or *approach 5* is the best choice for the monitoring systems developed in References [19,20]. *approach 1* might be the best choice to segment the reflected laser lines in good quality images. From the quantitative results, we could conclude that the recently proposed image processing algorithms are the most effective steps to form effective segmentation approaches. These image processing algorithms are all proposed based on the analysis of the modeling of the image intensity distribution.

**Table 1.** Comparison of computation accuracy for eight methods.

| Approaches | F-Measure |
| --- | --- |
| Approach 1 | 0.5380 |
| Approach 2 | 0.2074 |
| Approach 3 | 0.2276 |
| Approach 4 | 0.5011 |
| Approach 5 | 0.8546 |
| Approach 6 | 0.4743 |
| Approach 7 | 0.4469 |
| Approach 8 | 0.9176 |

**Table 2.** Comparison of computation time for eight methods.

| Approaches | Computation Time |
| --- | --- |
| Approach 7 | 0.05 s |
| Approach 3 | 0.045 s |
| Approach 2 | 0.042 s |
| Approach 8 | 0.031 s |
| Approach 6 | 0.028 s |
| Approach 5 | 0.0145 s |
| Approach 4 | 0.0138 s |
| Approach 1 | 0.01 s |

*6.2. Discussion*

The major contributions of the work include:

(1) We combine the recently proposed image processing algorithms and the traditional GLCM to propose different approaches to segment the reflected laser lines. Their performances including accuracy and processing time are evaluated and compared thoroughly in this paper, which is critical in implementing the on-line weld pool monitoring system;

(2) The image processing algorithms proposed previously are explained theoretically in this paper, which serves as a complementation to the previous research [23];

(3) More efficient segmentation approaches for images captured under mild welding parameters with relatively high quality are proposed in this paper.

Image segmentation is fundamental and challenging in many machine vision applications. The most effective methods are usually obtained from the formulated mathematical model and address the characteristics of the captured image sequences well. As a result, the best solution usually needs to combine different image processing algorithms and forms a heuristic approach to achieve the best accuracy and required efficiency. As a typical example of visual intelligent sensing, the research conducted in this work might benefit other researches that need to automatically and robustly extract visual information from the image sequences in different industrial applications.

**7. Conclusions**

For the image segmentation of the reflected laser lines during on line monitoring of weld pool surface, both accuracy and the efficiency are important. In this paper, we propose eight approaches to segment the reflected laser lines by combining different image processing algorithms. We evaluate their accuracy and efficiency extensively with both qualitative and quantitative results. Experimental results ranked the accuracy and the efficiency of the proposed approaches objectively. The quantitative results showed that the recently proposed image processing methods, including the difference method, the threshold selection method, the gradient detection method and the spline fitting method, are the most effective steps to form the effective segmentation approaches. The quality of the captured image is mainly determined by the welding process. During monitoring weld pool with violent changes, e.g.,

GMAW weld pool, all these recently proposed image processing methods should be combined as an approach to achieve the required accuracy. During monitoring gently changing weld pool, e.g., GTAW weld pool, only the difference method, the gradient detection method and the threshold selection method are required to form approach 5 that could meet the required accuracy while achieving higher efficiency.

**Author Contributions:** Conceptualization, Z.W. (Zhenzhou Wang); Methodology, Z.W. (Zhenzhou Wang); Software, Z.P. and Z.W. (Zihao Wang); Validation, X.Q., L.L. and J.P.; Formal Analysis, C.Z. and S.M.; Investigation, Z.W. (Zhenzhou Wang); Resources, Z.W. (Zhenzhou Wang); Data Curation, Z.W. (Zhenzhou Wang); Writing-Original Draft Preparation, Z.W. (Zhenzhou Wang); Writing-Review & Editing, C.Z., Z.P., Z.W. (Zihao Wang), X.Q, L.L., J.P. and S.M.; Visualization, Z.W. (Zhenzhou Wang); Supervision, Z.W. (Zhenzhou Wang); Project Administration, Z.W. (Zhenzhou Wang); Funding Acquisition, Z.W. (Zhenzhou Wang).

**Conflicts of Interest:** The authors declare no conflict of interest.

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
