# Peer review of "Image Segmentation Approaches for Weld Pool Monitoring during Robotic Arc Welding"

_applsci, doi:10.3390/app8122445_

Reviewer 1 Report

proposed approaches need more supporting example of weld pool scenario to understand in details

Author Response

proposed approaches need more supporting example of weld pool scenario to understand in details

Answer: The following comments are added and marked in yellow in the revision. Thanks.

Although the same weld pool monitoring system has been adopted in [9, 15-23], the monitoring algorithms have been proposed differently and divergently. In addition, most proposed monitoring approaches only work for the specifically designed welding scenario. As described in Section 3, either the proposed monitoring approach in [19] or the proposed monitoring approach in [20] could not work for the weld pool scenario in [23]. On the contrary, the proposed monitoring approach in [23] works better for the weld pool scenario in both [19] and [20] than their proposed monitoring approaches. The reason lies in that the quality of the captured laser line images in [19] and [20] is much higher than that of the captured images in [23]. When the quality of the captured image is reduced greatly, the requirement for the monitoring algorithms increases significantly. The proposed monitoring approach is the most robust one for the structured laser line based weld pool monitoring systems shown in Fig. 1 up to date. However, it might be redundant for monitoring of simple weld pool scenarios as described in [19] and [20]. In [19], GTAW process is used. The wire feed speed is 55 mm/s, the welding speed is 5 mm/s, the peak current is 220A and the base current is 50 A. In [20], GMAW-P process is used. The welding speed is 0 mm/s, the peak current is 160A and the base current is 80 A. In [23], GMAW-P process is used. The wire feed speed is 84.67 mm/s, the welding speed is 3.33 mm/s, the peak current is 230A and the base current is 70 A. As a result, the produced weld pools in [19-20] are much more stable than that produced in [23]. Thus, the quality of the captured images in [19-20] is much better than that of the images captured in [23]. On the other hand, the processing time is also important for on line monitoring. Therefore, the combination approach proposed in [23] is not always optimum when both accuracy and efficiency are considered. The monitoring approach should be designed as fast as possible after the accuracy has been met.

Reviewer 2 Report

This article focuses on image segmentation approaches for weld pool monitoring. The authors present a very interesting study concerning the determination of the most effective segmentation solutions for the weld pool images captured with different welding parameters. The proposed approaches are compared both qualitatively and quantitatively in this paper.  In Introduction section the authors justify the significance of investigations and their importance to monitor the  quality of robotic arc welding. The manuscript demonstrates that the authors understand relevant literatures enough in the field of weld pool monitoring. Both the quality of papers cited and number of references are suitable for Applied Sciences journal.

The paper contains the results of experimental and numerical investigations. The used experimental methods and numerical models are suitable, and the presentation of the results is very clear. In general, the work may be considered original and the results are sound. The structure of this paper and the quality of figures are perfect.

I would like to strongly recommend this paper for publication in the "Applied Sciences" journal after solving following observations:

-        The meaning of parameter f in Eq. (1) should be explained.

-        Line 197. Reference list does not consist reference [255].

-        Figure 2 should not be presented on two separate pages. Please reformat the text or divide this figure into two separate figures. See also Figs. 5, 8.

-        The caption of the figure 11 should be on the same page as the figure.

-        Line 436: solusion (?)

-        line 436: heruristic (?)

-        Conclusions are too general. This section should correspond with the main results obtained.

-        The style used in Reference List does not strictly correspond with Instructions for Authors.

Author Response

This article focuses on image segmentation approaches for weld pool monitoring. The authors present a very interesting study concerning the determination of the most effective segmentation solutions for the weld pool images captured with different welding parameters. The proposed approaches are compared both qualitatively and quantitatively in this paper.  In Introduction section the authors justify the significance of investigations and their importance to monitor the  quality of robotic arc welding. The manuscript demonstrates that the authors understand relevant literatures enough in the field of weld pool monitoring. Both the quality of papers cited and number of references are suitable for Applied Sciences journal.

Answer: Thanks much.

The paper contains the results of experimental and numerical investigations. The used experimental methods and numerical models are suitable, and the presentation of the results is very clear. In general, the work may be considered original and the results are sound. The structure of this paper and the quality of figures are perfect.

I would like to strongly recommend this paper for publication in the "Applied Sciences" journal after solving following observations:

Answer: Thanks much. All the following observations are solved and marked in yellow in the revision. 

-        The meaning of parameter f in Eq. (1) should be explained.

Answer: Explained.

-        Line 197. Reference list does not consist reference [255].

Answer: Corrected.

-        Figure 2 should not be presented on two separate pages. Please reformat the text or divide this figure into two separate figures. See also Figs. 5, 8.

Answer: Corrected.

-        The caption of the figure 11 should be on the same page as the figure.

Answer: Corrected.

-        Line 436: solusion (?)

Answer: Corrected.

-        line 436: heruristic (?)

Answer: Corrected.

-        Conclusions are too general. This section should correspond with the main results obtained.

Answer: Improved.

-        The style used in Reference List does not strictly correspond with Instructions for Authors

Answer: Corrected.

Round  2

Reviewer 1 Report

Implemetation of equations in this paper not explained in details  which need to improved.'

Author Response

Implemetation of equations in this paper not explained in details  which need to improved.

Answer: The implementations of equations are explained in Section 5 and marked in yellow.